# Body size and composition and risk of site-specific cancers in the UK Biobank and large international consortia: A mendelian randomisation study

**Mathew Vithayathil**[1], **Paul Carter**[2], **Siddhartha Kar**[2,3], **Amy M. Mason**[2], **Stephen Burgess**[2,4‡]*, **Susanna C. Larsson**[5,6‡]

**1** MRC Cancer Unit, University of Cambridge, Cambridge, United Kingdom, **2** Department of Public Health and Primary Care, University of Cambridge, Cambridge, United Kingdom, **3** MRC Integrative Epidemiology Unit, Bristol Medical School, University of Bristol, Bristol, United Kingdom, **4** MRC Biostatistics Unit, University of Cambridge, Cambridge, United Kingdom, **5** Unit of Cardiovascular and Nutritional Epidemiology, Institute of Environmental Medicine, Karolinska Institutet, Stockholm, Sweden, **6** Department of Surgical Sciences, Uppsala University, Uppsala, Sweden

‡ SB and SCL are joint last authors of this work.
* sb452@medschl.cam.ac.uk

**Data Availability Statement:** All primary data are available from the UK Biobank on application to any bona fide researcher (url: https://www.ukbiobank.

## Abstract

### Background

Evidence for the impact of body size and composition on cancer risk is limited. This mendelian randomisation (MR) study investigates evidence supporting causal relationships of body mass index (BMI), fat mass index (FMI), fat-free mass index (FFMI), and height with cancer risk.

### Methods and findings

Single nucleotide polymorphisms (SNPs) were used as instrumental variables for BMI (312 SNPs), FMI (577 SNPs), FFMI (577 SNPs), and height (293 SNPs). Associations of the genetic variants with 22 site-specific cancers and overall cancer were estimated in 367,561 individuals from the UK Biobank (UKBB) and with lung, breast, ovarian, uterine, and prostate cancer in large international consortia. In the UKBB, genetically predicted BMI was positively associated with overall cancer (odds ratio [OR] per 1 kg/m² increase 1.01, 95% confidence interval [CI] 1.00–1.02; $p = 0.043$); several digestive system cancers: stomach (OR 1.13, 95% CI 1.06–1.21; $p < 0.001$), esophagus (OR 1.10, 95% CI 1.03, 1.17; $p = 0.003$), liver (OR 1.13, 95% CI 1.03–1.25; $p = 0.012$), and pancreas (OR 1.06, 95% CI 1.01–1.12; $p = 0.016$); and lung cancer (OR 1.08, 95% CI 1.04–1.12; $p < 0.001$). For sex-specific cancers, genetically predicted elevated BMI was associated with an increased risk of uterine cancer (OR 1.10, 95% CI 1.05–1.15; $p < 0.001$) and with a lower risk of prostate cancer (OR 0.97, 95% CI 0.94–0.99; $p = 0.009$). When dividing cancers into digestive system versus non-digestive system, genetically predicted BMI was positively associated with digestive system cancers (OR 1.04, 95% CI 1.02–1.06; $p < 0.001$) but not with non-digestive system

ac.uk/). Genetic associations with cancer outcomes have been deposited at https://figshare.com/articles/dataset/Genetic_associations_with_cancer_outcomes/14806638.

**Funding:** Stephen Burgess is supported by Sir Henry Dale Fellowship jointly funded by the Wellcome Trust and the Royal Society (Grant Number 204623/Z/16/Z). This work was supported by the UK National Institute for Health Research Cambridge Biomedical Research Centre (BRC-1215-20014). The funders had no role in study design, data collection and analysis, decision to publish, or preparation of the manuscript.

**Competing interests:** The authors have declared that no competing interests exist apart from the following: Stephen Burgess is a paid statistical reviewer for PLOS Medicine.

**Abbreviations:** BCAC, Breast Cancer Association Consortium; BMI, body mass index; CI, confidence interval; ER−, estrogen receptor negative; ER+, estrogen receptor positive; FFMI, fat-free mass index; FMI, fat mass index; GWAS, genome-wide association study; HES, hospital episodes statistics; ICD, International Classification of Diseases; IGF1, insulin-like growth factor 1; MR, mendelian randomisation; OR, odds ratio; SNP, single nucleotide polymorphism; STROBE, Strengthening the Reporting of Observational Studies in Epidemiology; UKBB, UK Biobank.

cancers (OR 1.01, 95% CI 0.99–1.02; $p = 0.369$). Genetically predicted FMI was positively associated with liver, pancreatic, and lung cancer and inversely associated with melanoma and prostate cancer. Genetically predicted FFMI was positively associated with non-Hodgkin lymphoma and melanoma. Genetically predicted height was associated with increased risk of overall cancer (OR per 1 standard deviation increase 1.09; 95% CI 1.05–1.12; $p < 0.001$) and multiple site-specific cancers. Similar results were observed in analyses using the weighted median and MR–Egger methods. Results based on consortium data confirmed the positive associations between BMI and lung and uterine cancer risk as well as the inverse association between BMI and prostate cancer, and, additionally, showed an inverse association between genetically predicted BMI and breast cancer. The main limitations are the assumption that genetic associations with cancer outcomes are mediated via the proposed risk factors and that estimates for some lower frequency cancer types are subject to low precision.

## Conclusions

Our results show that the evidence for BMI as a causal risk factor for cancer is mixed. We find that BMI has a consistent causal role in increasing risk of digestive system cancers and a role for sex-specific cancers with inconsistent directions of effect. In contrast, increased height appears to have a consistent risk-increasing effect on overall and site-specific cancers.

## Author summary

### Why was this study done?

- The causal relevance of body size and composition as risk factors for specific cancers is unclear based on traditional observational studies.

- By considering the relationships between genetically predicted values of body size and composition with cancer risk, our estimates are less influenced by confounding variables, and, hence, more reliably reflect the underlying causal relationships between these measures and cancer risk.

### What did the researchers do and find?

- We assessed the associations between genetically predicted body mass index (BMI), fat mass index (FMI), fat-free mass index (FFMI), and height with 22 specific cancers in the UK Biobank (UKBB), a population-based sample of the United Kingdom residents.

- Although genetically predicted height was consistently associated with increased risk of site-specific cancers, genetically predicted BMI was associated with an increased risk of certain digestive system cancers (esophageal, stomach, liver, and pancreas), plus lung and uterine cancer, but a decreased risk of breast and prostate cancer.

- When dividing cancers into digestive system cancers versus non-digestive system cancers, genetically predicted BMI was associated with increased risk of digestive system cancers, but not associated with non-digestive system cancers.

### What do these findings mean?

- Our findings suggest that BMI is a causal risk factor for some cancers, but is not a generic risk factor for all cancers.

- Body fat may play a role in development of specific cancers and should be studied further to identify future targets to prevent cancer.

- Public health strategies should focus on reducing obesity as a risk factor for cancer, but should be clear that benefit may be limited to certain cancers.

## Background

Obesity is a global epidemic [1] that is predicted to affect 20% of the world's population by 2025 [2]. The relationship between obesity and cancer risk has been subject to extensive investigation. Body mass index (BMI) is the most commonly measured marker for obesity and correlates most strongly with fat mass [3]. Observational studies have shown that raised BMI is associated with increased risk [4–7], no risk [4,7], and even reduced risk [4,8] of cancers. In particular, consistent positive associations have been observed between BMI and risk of colorectal, stomach, esophagus, liver, gallbladder, breast, endometrial, ovarian, kidney, and pancreatic cancers [9–11]. However, traditional epidemiological studies are influenced by confounding factors, such as smoking [12,13], and reverse causation due to subclinical disease [14,15]. Thus, the true relationship between obesity and cancer remains unclear.

Mendelian randomisation (MR) uses genetic variants as instrumental variables for an exposure to investigate evidence for a causal effect of the exposure on a disease outcome [16]. MR estimates represent associations of genetically predicted levels of risk factors with outcomes, as opposed to standard epidemiological estimates, which represent associations of the risk factor levels with outcomes. As a result of Mendel's laws of segregation and independent assortment, estimates from MR are less susceptible to bias due to confounding factors than those from conventional observational epidemiological studies [17]. As the genetic code cannot be influenced by environmental factors or preclinical disease, MR estimates are also less susceptible to bias due to reverse causation.

MR investigations have previously been performed to investigate the effect of obesity on various cancer types. Studies have suggested risk-increasing effects of BMI for cancers of the colorectum, pancreas, kidney, lung, uterus, and ovary, for adenocarcinoma in the esophagus, and for overall cancer [18–20]. A separate study evidenced a risk-increasing effect of BMI on renal cell carcinoma [21]. Risk-decreasing effects of BMI have been evidenced for both pre- and postmenopausal breast cancer [22] and for prostate cancer in one study [23], but not others [19,24]. However, a comprehensive MR investigation into the effect of BMI on a wide range of cancer types has not been performed. Additionally, previous investigations have not considered the specific contribution of fat mass and fat-free mass to cancer risk.

Our aim in this paper is to perform a wide-angled MR investigation to provide independent evidence that replicates and extends analyses for the impact of obesity on cancer outcomes

where MR investigations have already been performed and provides new evidence for cancer outcomes that have not previously been investigated using this approach. In particular, we want to assess the consistency of findings across different site-specific cancers. We define fat mass index (FMI) and fat-free mass index (FFMI) analogously to BMI, as fat mass or fat-free mass divided by height squared. As a comparative analysis, we consider height as a risk factor, as this has also been suggested to have a causal effect on multiple cancer types [25–27]. We used MR to investigate the causal roles of genetically predicted BMI, FMI, FFMI, and height on the risk of developing 22 cancers in 367,561 individuals from the UK Biobank (UKBB) study. We supplement our investigation with publicly available genetic association data from large international consortia for certain site-specific cancers. We aimed to elucidate the causal role of body composition for site-specific cancers and so extend and focus the evidence base for targeted public health strategies.

## Methods

### Study population

Data for genetic associations with site-specific cancer risk were obtained from the UKBB. The UKBB comprises demographic, clinical, biochemical, and genetic data from around 500,000 adults (aged 37 to 73 years old at baseline) recruited between 2006 and 2010 and followed up until June 30, 2020 [28]. Only unrelated individuals of European ancestries (defined by self-report and genetics) were included in our analysis in order to reduce population stratification bias. For each group of related individuals (third-degree relatives or closer), only 1 individual was included in analyses. After performing quality control filters as described previously [29], 367,561 individuals were included in analyses (S1 Fig). We defined cancer outcomes in the UKBB for the 22 most common site-specific cancers in the UK using the International Classification of Diseases (ICD)-9 and ICD-10 coding (S1 Table). Overall cancer analyses included individuals with any of the 22 site-specific cancers. Cancer outcomes were obtained from electronic health records, hospital episodes statistics (HES) data, the National Cancer Registry, death certification data, and self-reporting validated by nurse interview. Both prevalent and incident events were included in analyses. Genetic association estimates were obtained for each cancer outcome by logistic regression adjusting for age, sex, and 10 genetic principal components. Associations for sex-specific cancers (breast, ovarian, cervical, and uterine for women and testicular and prostate for men) were estimated in participants of the relevant sex only (198,825 women and 168,736 men).

In addition, publicly available data were extracted from the MR-Base platform [30] for lung, breast, ovarian, uterine, and prostate cancer from the International Lung Cancer Consortium (11,348 cases and 15,861 controls) [31], Breast Cancer Association Consortium (BCAC) (122,977 cases and 105,974 controls) [32], the Ovarian Cancer Association Consortium (25,509 cases and 40,941 controls) [33], a meta-analysis of genome-wide association studies (GWASs) for endometrial cancer (12,906 cases and 108,979 controls from 17 studies identified from the Endometrial Cancer Association Consortium, the Epidemiology of Endometrial Cancer Consortium, and the UKBB) [34], and the Prostate Cancer Association Group to Investigate Cancer Associated Alterations in the Genome consortium (79,148 cases and 61,106 controls) [35].

### Genetic instruments

The genetic instrument for BMI compromised 312 uncorrelated single nucleotide polymorphisms (SNPs) (linkage disequilibrium $R^2 < 0.001$) associated with BMI at genome-wide significance ($p < 5 \times 10^{-8}$) in a GWAS of up to 806,834 individuals of European ancestries [36]

(S2 Table). For fat mass and fat-free mass (measured using bioelectrical impedance), we used 577 uncorrelated SNPs associated with body composition among 331,291 UKBB participants [37] (S3 Table). FMI and FFMI were computed by dividing fat mass or fat-free mass by the square of height. A GWAS of 253,288 European ancestry individuals identified 697 genome-wide significant SNPs ($p < 5 \times 10^{-8}$) for adult height [38], of which 293 were uncorrelated (linkage disequilibrium $R^2 < 0.001$) and were used as instrumental variables (S4 Table). Genetic associations with these body composition measures were obtained from the relevant discovery GWAS and were adjusted for age, sex, and genetic principal components.

## Statistical analysis

Associations of genetically predicted BMI and height with the 22 site-specific cancers and overall cancer for the UKBB cohort were obtained using the random-effects inverse-variance weighted method [39]. We performed additional analyses using the weighted median [39] and MR–Egger [40] methods. The analyses of FMI and FFMI were based on the multivariable random-effects inverse-variance weighted method with both exposures included in the same model. Although overall cancer is a composite outcome comprising malignancies with different etiologies, analyses for overall cancer are important from a public health perspective to estimate the overall impact of the risk factors on cancer risk.

Our analysis did not have an explicit prespecified analysis plan. The analysis was conducted similarly to previous published efforts for investigating the causal relationships of BMI, FMI, and FFMI with cardiovascular diseases [37,41]. In response to comments from peer reviewers, we made a number of changes to the analysis: We changed the overall cancer outcome from including all cancers to including any of the 22 named site-specific cancers, we updated the genetic variants used as instrumental variables to those from the latest GWAS for the relevant risk factor, and we added analyses based on large-scale consortia for lung, breast, ovarian, uterine, and prostate cancer.

The odds ratios (ORs) are expressed per 1 kg/m$^2$ increase in genetically predicted BMI, FMI, and FFMI and per 1 standard deviation (approximately 6.5 cm) increase in height. As the number of cases and thus statistical power differed between analyses, we did not set a fixed threshold for statistical significance. Statistical analyses were performed in Stata/SE 14.2 using the mrrobust package [42] and R 3.6.0 software using the MendelianRandomization package [39]. Power calculations were performed using the web-based tool at https://sb452.shinyapps.io/power/ [43].

This study is reported as per the Strengthening the Reporting of Observational Studies in Epidemiology (STROBE) guideline (S1 Checklist).

## Results

Baseline characteristics of the 367,561 participants in the UKBB are provided in Table 1. The mean age was 57.2 years, with 45.9% males. Moreover, 10.3% of participants were current smokers. In the UKBB, the 312 SNPs explained 4.1% of the variance in BMI, whereas the 577 SNPs for body composition explained 3.1% of the variance in FMI and 2.3% of the variance in FFMI. The 293 SNPs for height explained around 5.5% of the variance in height. The phenotypic correlation of BMI with FMI was 0.84 and with FFMI was 0.66. The correlation between FMI and FFMI was 0.14. The relatively low correlation between FMI and FFMI means multivariable analyses can likely differentiate between these 2 risk factors. A total of 59,647 participants had one of the 22 defined site-specific cancers. When excluding outcomes for which only self-reported data were available, 55 674 events remained.

**Table 1. Baseline characteristics of the UKBB participants included in this study.**

|  | Mean (SD) or $n$ (%)[†] |
|---|---|
| **Sample size** | 367,561 (100) |
| **Male** | 168,736 (45.9) |
| **Age at baseline (years)** | 57.2 (8.1) |
| **Body composition** |  |
| BMI (kg/m$^2$) | 27.4 (4.8) |
| FMI (kg/m$^2$) | 8.8 (3.6) |
| FFMI (kg/m$^2$) | 18.6 (2.6) |
| Height (m) | 1.69 (0.07) |
| **Blood pressure (mm Hg)** |  |
| Systolic blood pressure | 137.6 (18.6) |
| Diastolic blood pressure | 82.0 (10.1) |
| **Smoking status** |  |
| Current | 37,866 (10.3) |
| Ex | 185,704 (50.5) |
| Never | 143,777 (39.1) |
| **Alcohol status** |  |
| Current | 342,797 (93.2) |
| Ex | 12,732 (3.5) |
| Never | 11,646 (3.2) |
| **Type 2 diabetes** | 15,834 (4.3) |

[†] Mean (standard deviation) for continuous variables; $n$ (%) for categorical variables.

A total of 214 participants had missing data on smoking status, and 386 participants had missing data on alcohol status.

BMI, body mass index; FFMI, fat-free mass index; FMI, fat mass index; SD, standard deviation; UKBB, UK Biobank.

Power calculations are provided in S2 Fig. For BMI, there was 80% power to detect an OR of 1.2 per 1 kg/m$^2$ increase in genetically predicted BMI even for outcomes with only 300 events and an OR of 1.05 for outcomes with 3,800 events. Power was lower for FMI and FFMI, although still adequate to detect a moderate effect size for more common cancers.

## BMI and cancer risk

Associations between genetically predicted BMI and site-specific and overall cancer in the UKBB are shown in Fig 1. Genetically predicted BMI was associated with an increased risk of overall cancer (OR 1.01, 95% confidence interval [CI] 1.00 to 1.02; $p = 0.043$); certain digestive system cancers, including stomach (OR 1.13, 95% CI 1.06 to 1.21; $p < 0.001$), esophagus (OR 1.10, 95% CI 1.10 (1.03, 1.17; $p = 0.003$), liver (OR 1.13, 95% CI 1.03 to 1.25; $p = 0.012$), and pancreas (OR 1.06, 95% CI 1.01 to 1.12; $p = 0.016$); and lung cancer (OR 1.08, 95% CI 1.04 to 1.12; $p < 0.001$). For sex-specific cancers, elevated BMI was associated with an increased risk of uterine cancer (OR 1.10, 95% CI 1.05 to 1.15; $p < 0.001$) and with a lower risk of prostate cancer (OR 0.97, 95% CI 0.94 to 0.99; $p = 0.009$). After omission of self-reported cancer events, the strength of association with risk of overall cancer was diminished (OR 1.01, 95% CI 1.00 to 1.02; $p = 0.055$). Results for site-specific cancers excluding self-reported outcomes were generally similar but marginally less precise (S3 Fig). Complementary analyses based on the weighted median and MR–Egger methods revealed similar but less precise estimates (S5 Table).

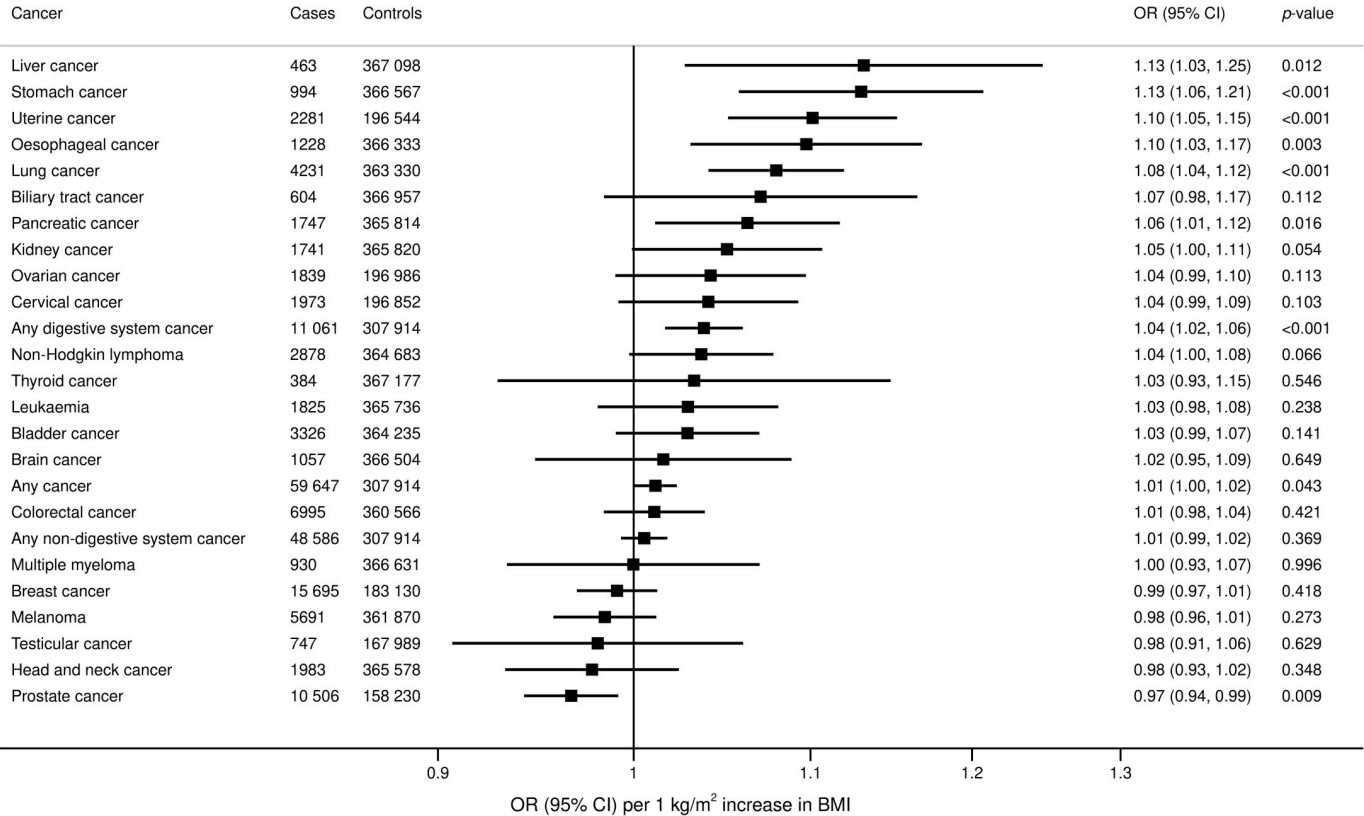

**Fig 1. Associations of genetically predicted BMI with overall and site-specific cancers in the UKBB.** ORs are expressed per 1 kg/m$^2$ increase in BMI. Results are obtained from the random-effects inverse-variance weighted method. BMI, body mass index; CI, confidence interval; OR, odds ratio; UKBB, UK Biobank.

In analyses based on consortium data, genetically predicted BMI was positively associated with increased risk of overall lung cancer (OR 1.04, 95% CI 1.01 to 1.07; $p = 0.006$) and the squamous cell cancer subtype (OR 1.08, 95% CI 1.03 to 1.12; $p < 0.001$) (Fig 2). For sex-specific cancers, elevated BMI was associated with increased risk of uterine cancer (OR 1.13, 95% CI 1.10 to 1.16; $p < 0.001$), but with lower risk of breast (OR 0.96, 95% CI 0.95 to 0.98; $p < 0.001$) and prostate cancer (OR 0.98, 95% CI 0.96 to 1.00; $p = 0.020$). Inverse associations with similar magnitude were observed for both estrogen receptor positive (ER+) and estrogen receptor negative (ER−) breast cancer. Positive associations were observed for mucinous and endometrioid ovarian cancer, but not overall ovarian cancer.

### FMI and FFMI and cancer risk

Similar to BMI, genetically predicted FMI was associated with an increased risk of liver (OR 2.40, 95% CI 1.02 to 5.65; $p = 0.045$), pancreatic (OR 1.68, 95% CI 1.06 to 2.66; $p = 0.028$), and lung cancer (OR 1.57, 95% CI 1.16 to 2.13; $p = 0.004$) in the UKBB (Fig 3). Elevated FMI was associated with lower risk of prostate cancer (OR 0.77, 95% CI 0.61 to 0.97; $p = 0.027$) and melanoma (OR 0.68, 95% CI 0.51 to 0.91; $p = 0.008$). From the consortia, elevated FMI was associated with increased uterine cancer risk (OR 1.69, 95% CI 1.33 to 2.15; $p < 0.001$) (Fig 4).

Genetically predicted FFMI was associated with increased risk of non-Hodgkin lymphoma (OR 1.89, 95% CI 1.21 to 2.96; $p = 0.005$) and melanoma (OR 1.44, 95% CI 1.02 to 2.05; $p = 0.039$) in the UKBB (Fig 3). From the consortia, elevated FFMI was associated with a decreased risk of breast cancer (0.77, 95% CI 0.64 to 0.93; $p = 0.007$) (Fig 4).

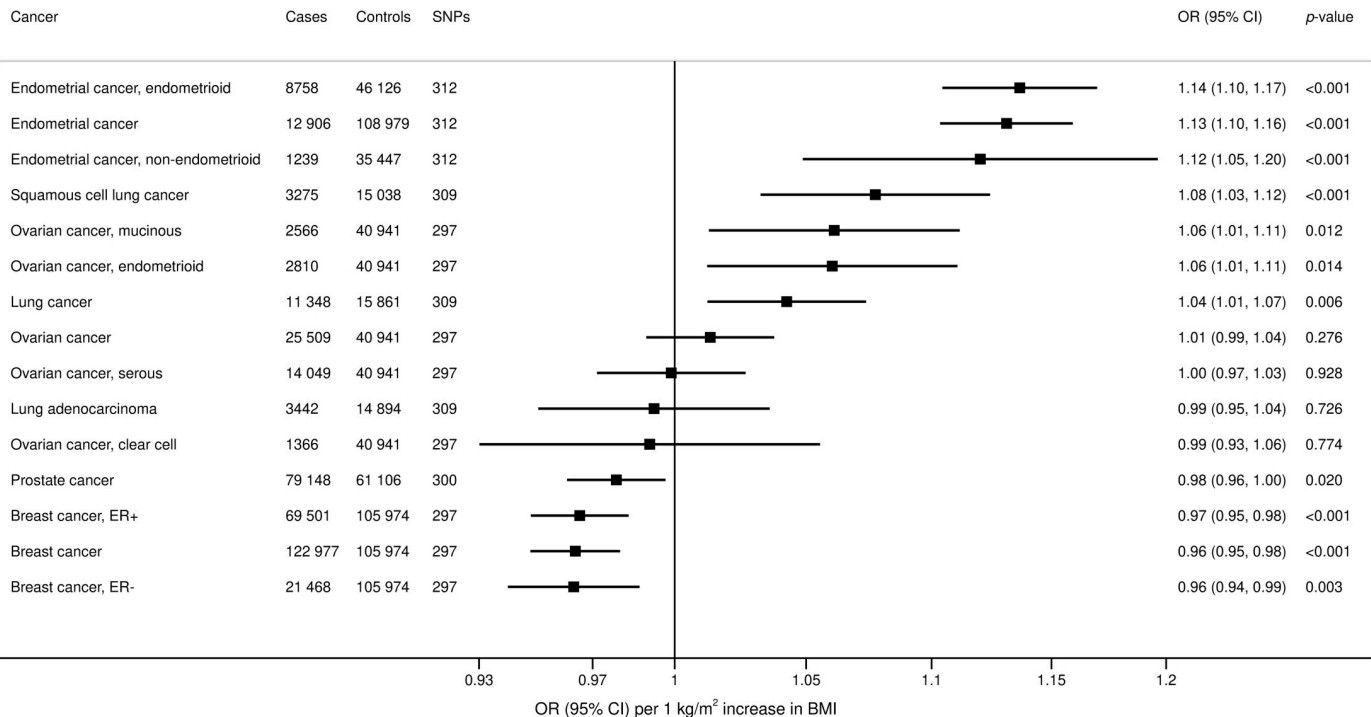

**Fig 2. Associations of genetically predicted BMI with site-specific cancers in large international consortia.** ORs are expressed per 1 kg/m² increase in BMI. Results are obtained from the random-effects inverse-variance weighted method. BMI, body mass index; CI, confidence interval; ER−, estrogen receptor negative; ER+, estrogen receptor positive; OR, odds ratio; SNP, single nucleotide polymorphism.

## Height and cancer risk

In the UKBB, genetically predicted height was positively associated with overall cancer (OR 1.09; 95% 1.05 to 1.12; $p < 0.001$) and multiple site-specific cancers, including kidney (OR 1.19, 95% CI 1.02 to 1.38; $p = 0.027$), colorectal (OR 1.09, 95% CI 1.01 to 1.19; $p = 0.034$), biliary tract (OR 1.45, 95% CI 1.14 to 1.84; $p = 0.003$), breast (OR 1.12, 95% CI 1.06 to 1.19; $p < 0.001$), and ovarian cancer (OR 1.21, 95% CI 1.05 to 1.40; $p = 0.008$) (Fig 5). Estimates were generally similar in additional analyses using the MR–Egger and weighted median methods (S6 Table). From the consortia, genetically predicted height was positively associated with ovarian (OR 1.37, 95% CI 1.14 to 1.64; $p = 0.001$) and breast cancer (OR 1.10, 95% CI 1.05 to 1.15; $p < 0.001$), similar to the UKBB (Fig 6).

## Digestive system versus non-digestive system cancer risk

When dividing cancers into digestive system (esophagus, stomach, colorectum, liver, biliary tract, and pancreas; 11,061 cases) versus non-digestive system (48,586 cases), estimates for BMI were OR 1.04 (95% CI 1.02 to 1.06; $p < 0.001$) for digestive system and OR 1.01 (95% CI 0.99 to 1.02; $p = 0.37$) for non-digestive system cancers. For FMI, estimates were OR 1.17 (95% CI 0.95 to 1.44; $p = 0.13$) for digestive system and OR 0.99 (95% CI 0.88 to 1.11; $p = 0.86$) for non-digestive system cancers. For FFMI, estimates were OR 1.11 (95% CI 0.86 to 1.43; $p = 0.42$) for digestive system and OR 1.12 (95% CI 0.96 to 1.29; $p = 0.15$) for non-digestive system cancers. Genetically predicted height was positively associated with both digestive system and non-digestive system cancers (Fig 5).

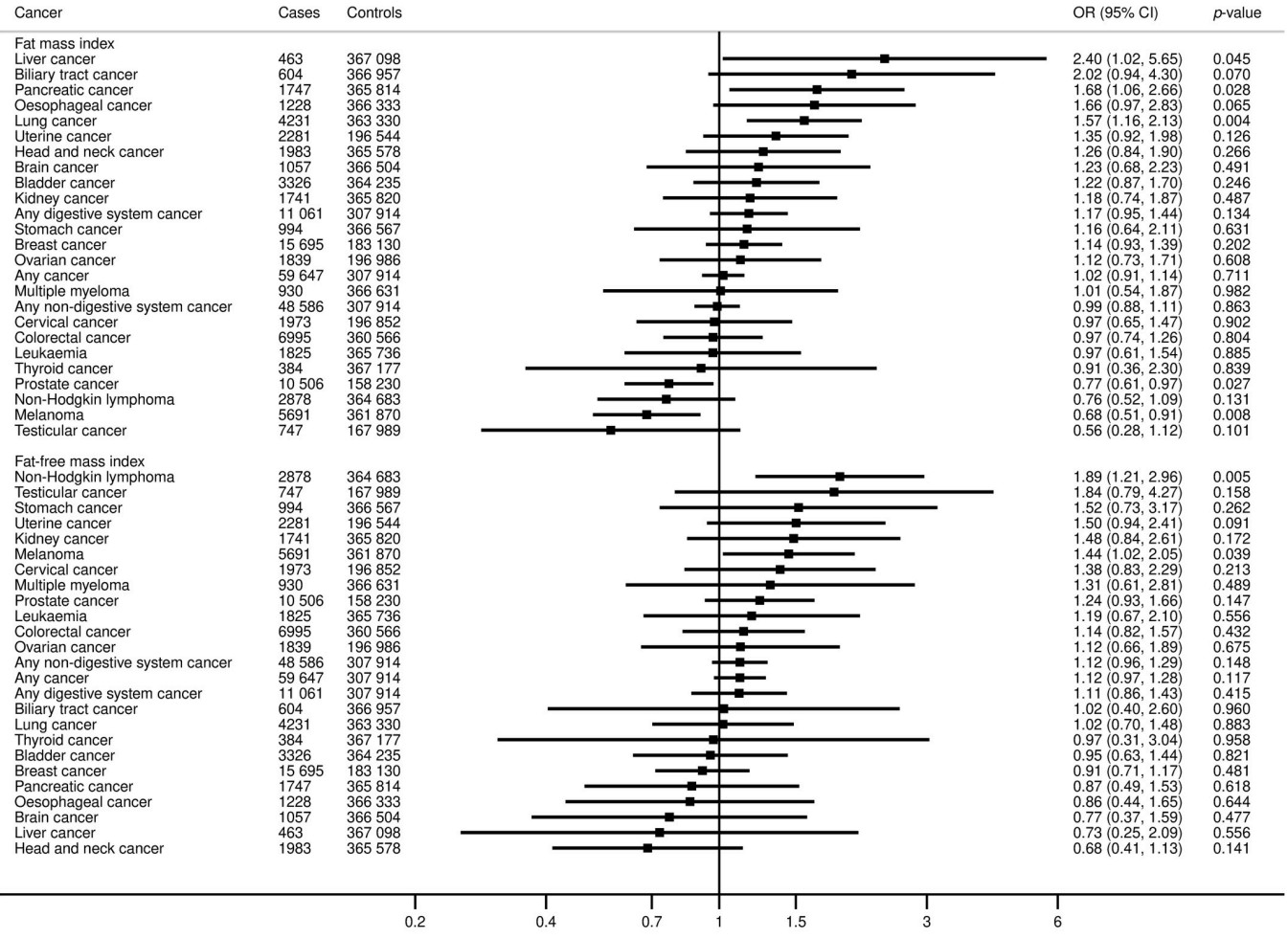

**Fig 3. Associations of genetically predicted FMI and FFMI with overall and site-specific cancers in the UKBB.** ORs are expressed per one 1 kg/m² increase in FMI. Results are obtained from the multivariable random-effects inverse-variance weighted method. CI, confidence interval; FFMI, fat-free mass index; FMI, fat mass index; OR, odds ratio; UKBB, UK Biobank.

## Discussion

This MR study investigated the causal role of clinically relevant measures of body composition for a wide range of site-specific cancers. Genetically predicted BMI was associated with risk of overall cancer. Elevated BMI was positively associated with several digestive system cancers, including at the esophagus, stomach, liver, and pancreas. Additionally, BMI was positively associated with lung and uterine cancer, but inversely associated with breast cancer (only in the BCAC) and prostate cancer. Genetically predicted FMI was positively associated with lung, liver, and pancreatic cancer, with inverse associations seen for melanoma and prostate cancer. FFMI was positively associated with non-Hodgkin lymphoma, melanoma, and uterine cancer, with inverse associations seen for breast cancer. Genetically predicted height was positively associated with overall and multiple site-specific cancers.

The relationship between adiposity and cancer has been assessed in many traditional observational studies. An umbrella review of 204 meta-analyses found adiposity to be consistently associated with 10 site-specific cancers: esophageal, gastric, colorectal, biliary tract, pancreas,

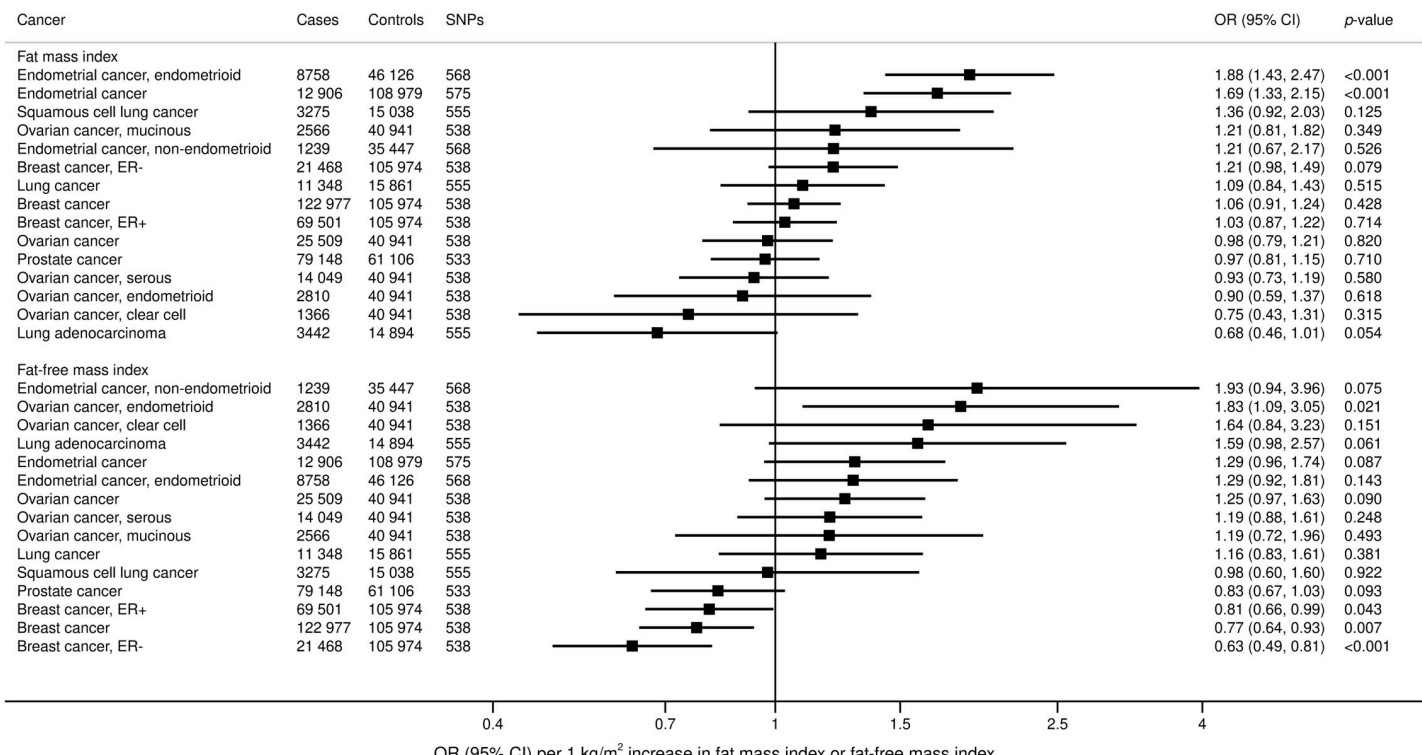

**Fig 4. Associations of genetically predicted FMI and FFMI with site-specific cancers in large international consortia.** ORs are expressed per one 1 kg/m² increase in FMI. Results are obtained from the multivariable random-effects inverse-variance weighted method. CI, confidence interval; ER−, estrogen receptor negative; ER+, estrogen receptor positive; FFMI, fat-free mass index; FMI, fat mass index; OR, odds ratio; SNP, single nucleotide polymorphism.

breast, uterine, ovarian, and kidney cancer as well as multiple myeloma [11]. Current MR studies have replicated the associations with esophageal, colorectal, pancreas, pancreas, uterine, ovarian, and kidney cancer [18–21,44,45]. However, further positive associations were observed in MR analyses for lung cancer [19,20], and inverse associations were seen with breast and prostate cancer [22,23]. These discrepancies between observational and MR studies may be due to the effect of environmental confounders (such as smoking) and reverse causation bias in traditional observational studies.

In our study, we observed a positive association between genetically predicted BMI and digestive system cancers. This link between BMI and adiposity with risk of certain digestive system cancers replicates and extends previous findings. High BMI has been associated with esophageal and stomach cancers in a meta-analysis of observational studies [46] and MR studies [20]. Increased adipose tissue is associated with insulin resistance and hyperinsulinemia [47], with raised circulating insulin enhancing colorectal epithelial cell proliferation in rat models [48]. Additionally, ghrelin is a gut hormone produced in the stomach, with reduced levels seen in obesity [49]. Ghrelin reduces pro-inflammatory cytokines and inflammatory stress [50], and reduced levels are associated with increased risk of esophageal [51] and stomach [52] cancers. Adiposity is also well established in causing nonalcoholic fatty liver disease and has been implicated in its progression to hepatocellular carcinoma [53]. In line with this, we observed an increased risk of liver cancer with raised BMI and FMI. Similarly, we observed a positive association between elevated BMI and FMI and pancreas cancer, in line with adipose tissue driving low grade inflammation and carcinogenesis in pancreatic tissue through pro-inflammatory cytokines [54]. We also observe low-precision positive associations between

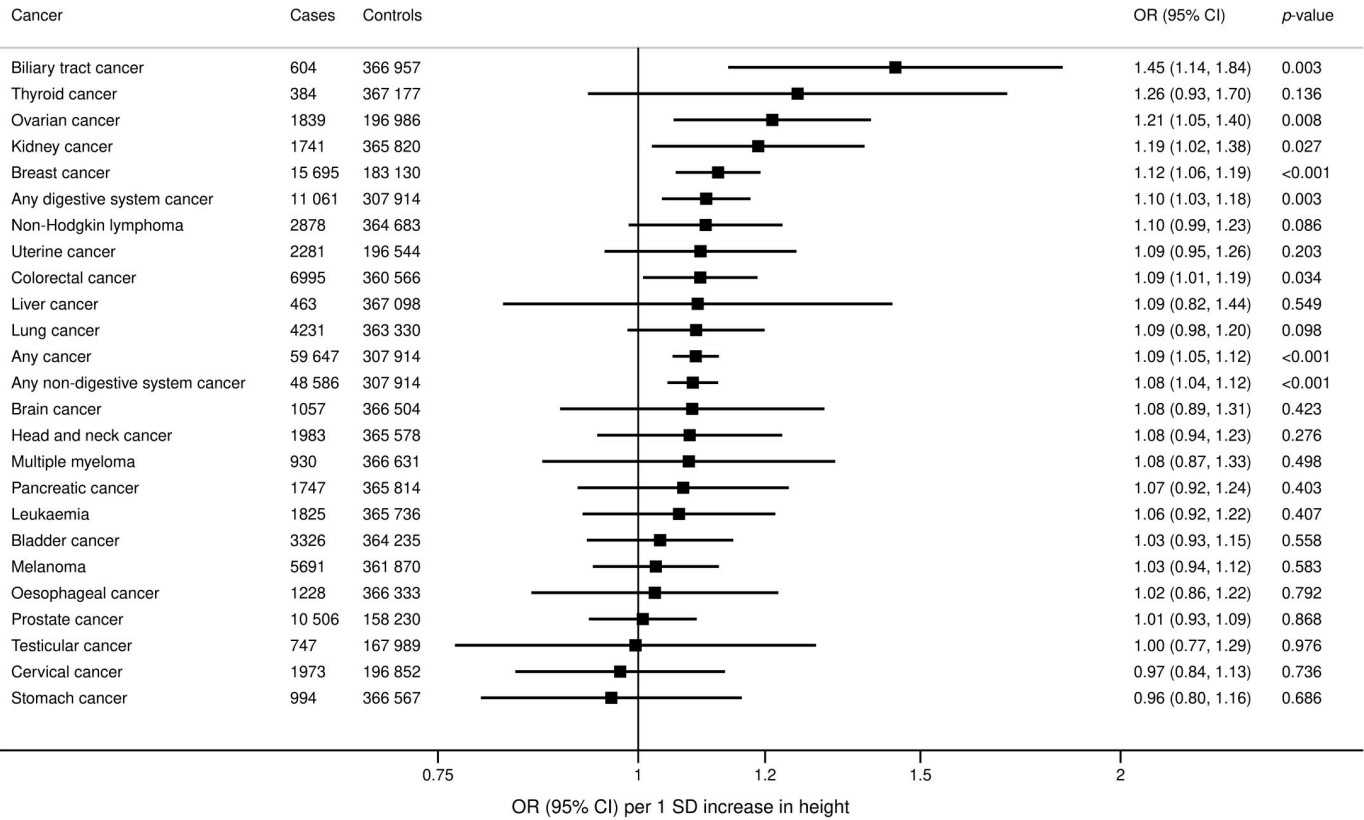

| Cancer | Cases | Controls | OR (95% CI) | p-value |
|---|---|---|---|---|
| Biliary tract cancer | 604 | 366 957 | 1.45 (1.14, 1.84) | 0.003 |
| Thyroid cancer | 384 | 367 177 | 1.26 (0.93, 1.70) | 0.136 |
| Ovarian cancer | 1839 | 196 986 | 1.21 (1.05, 1.40) | 0.008 |
| Kidney cancer | 1741 | 365 820 | 1.19 (1.02, 1.38) | 0.027 |
| Breast cancer | 15 695 | 183 130 | 1.12 (1.06, 1.19) | <0.001 |
| Any digestive system cancer | 11 061 | 307 914 | 1.10 (1.03, 1.18) | 0.003 |
| Non-Hodgkin lymphoma | 2878 | 364 683 | 1.10 (0.99, 1.23) | 0.086 |
| Uterine cancer | 2281 | 196 544 | 1.09 (0.95, 1.26) | 0.203 |
| Colorectal cancer | 6995 | 360 566 | 1.09 (1.01, 1.19) | 0.034 |
| Liver cancer | 463 | 367 098 | 1.09 (0.82, 1.44) | 0.549 |
| Lung cancer | 4231 | 363 330 | 1.09 (0.98, 1.20) | 0.098 |
| Any cancer | 59 647 | 307 914 | 1.09 (1.05, 1.12) | <0.001 |
| Any non-digestive system cancer | 48 586 | 307 914 | 1.08 (1.04, 1.12) | <0.001 |
| Brain cancer | 1057 | 366 504 | 1.08 (0.89, 1.31) | 0.423 |
| Head and neck cancer | 1983 | 365 578 | 1.08 (0.94, 1.23) | 0.276 |
| Multiple myeloma | 930 | 366 631 | 1.08 (0.87, 1.33) | 0.498 |
| Pancreatic cancer | 1747 | 365 814 | 1.07 (0.92, 1.24) | 0.403 |
| Leukaemia | 1825 | 365 736 | 1.06 (0.92, 1.22) | 0.407 |
| Bladder cancer | 3326 | 364 235 | 1.03 (0.93, 1.15) | 0.558 |
| Melanoma | 5691 | 361 870 | 1.03 (0.94, 1.12) | 0.583 |
| Oesophageal cancer | 1228 | 366 333 | 1.02 (0.86, 1.22) | 0.792 |
| Prostate cancer | 10 506 | 158 230 | 1.01 (0.93, 1.09) | 0.868 |
| Testicular cancer | 747 | 167 989 | 1.00 (0.77, 1.29) | 0.976 |
| Cervical cancer | 1973 | 196 852 | 0.97 (0.84, 1.13) | 0.736 |
| Stomach cancer | 994 | 366 567 | 0.96 (0.80, 1.16) | 0.686 |

OR (95% CI) per 1 SD increase in height

**Fig 5. Associations of genetically predicted height with overall and site-specific cancers in the UKBB.** ORs are expressed per 1 standard deviation (6.5 cm) increase in height. Results are obtained from the random-effects inverse-variance weighted method. CI, confidence interval; OR, odds ratio; UKBB, UK Biobank.

elevated BMI and colorectal and biliary tract cancers, suggesting a causal role between adiposity and digestive system cancers. Further research should assess the impact of weight loss interventions and dietary interventions in reducing cancers of the digestive system.

In this MR study, elevated genetically predicted BMI was associated with sex-specific cancers: increased risk of uterine cancer and decreased risk of breast and prostate cancer. BMI and breast cancer has been extensively studied in previous MR studies [19,22,55,56]. A previous MR study based on data from the BCAC and DRIVE consortia of 46,325 cases of breast cancer found that genetically predicted BMI based on 84 SNPs was inversely associated with breast cancer risk in both pre- and postmenopausal women [22], consistent with the findings of our study based on data from the BCAC. A further MR study of 98,842 cases of breast cancer confirmed this finding [57]. These MR results contrast with observational study findings that have demonstrated adult obesity to be associated with increased risk of postmenopausal breast cancer [10,46]. Furthermore, our study corroborates the findings of a previous MR study of 6,609 cases of uterine cancer, showing genetically predicted BMI was positively associated with incidence [58]. In males, our findings are in line with a previous MR of 22 European cohorts, which showed weak evidence for an inverse association between genetically predicted BMI and prostate cancer [24], as well as a larger analysis showing stronger evidence for an association [23]. The association of BMI and these sex-specific cancers is likely to be at least in part hormonally mediated. In males, elevated BMI reduces serum testosterone [59], with androgens recognised as promoting prostate cancer. In premenopausal women, increased BMI is associated with anovulation, reducing lifetime exposure of circulating estrogen and progesterone

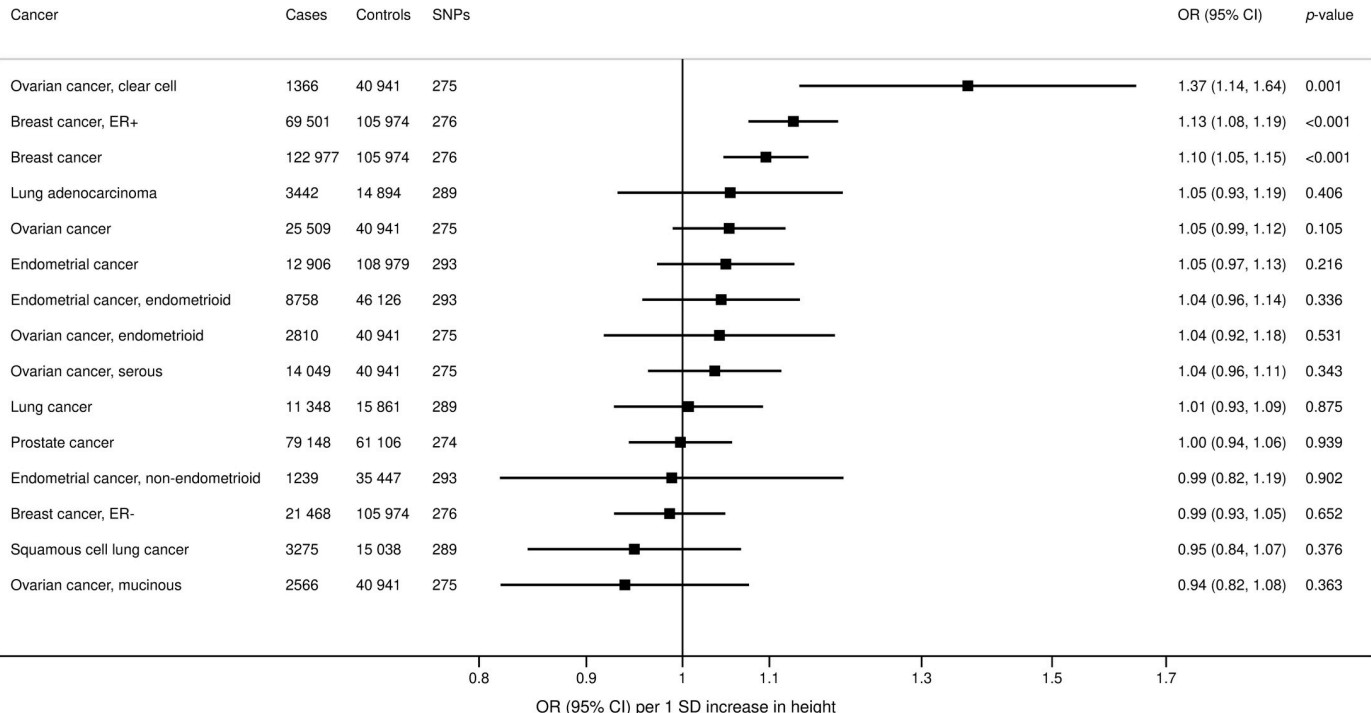

**Fig 6. Associations of genetically predicted height with site-specific cancers in large international consortia.** ORs are expressed per 1 standard deviation (6.5 cm) increase in height. Results are obtained from the random-effects inverse-variance weighted method. CI, confidence interval; ER−, estrogen receptor negative; ER+, estrogen receptor positive; OR, odds ratio; SNP, single nucleotide polymorphism.

[60], and thus a consequent reduction in breast cancer risk [61] and an increase in uterine cancer risk [62]. While observational associations of BMI with pre- and postmenopausal breast cancers are directionally discordant, MR estimates have consistent direction. This may be because the association of genetically predicted BMI is mediated via lifelong exposure to elevated estrogen levels, the majority of which is premenopausal. We report an inverse association between FFMI and breast cancer risk, suggesting that increased non-adipose tissue density may have a protective role against malignancy. Observational studies have demonstrated that sarcopenia is associated with increased breast cancer mortality [63,64], although the association may be subject to reverse causation. However, reduced muscle mass and atrophy is associated with systemic inflammation [65,66] and increased TNF-alpha levels [66], which may drive carcinogenesis, although the mechanistic links need to be assessed further.

We report a positive association between genetically predicted BMI and squamous cell lung cancer risk, consistent with previous MR findings [19]. Our findings of no association between BMI and melanoma risk are consistent with findings of a recent MR investigation [67]. However, we observed an inverse association between FMI and melanoma risk, but a positive association for FFMI. These findings suggest that body composition influences melanoma risk, with increased adiposity being protective against melanoma. This may be because those with higher FFMI and lower FMI spend more time outdoors and so have greater exposure to the sun. Previous observational studies have shown that obesity is associated with improved survival of melanoma patients [68], with a down-regulation of key lipid genes shown in melanoma cells [69], suggesting that lipids play a role in carcinogenesis.

We observed a positive association between genetically predicted height and overall cancer risk, which was consistent across a wide range of site-specific cancers, including kidney,

colorectal, biliary tract, ovarian, and breast cancers. Our findings are consistent with previous MR studies showing a positive association of height with colorectal [70–72] and breast cancer [72–74]. Increased height is associated with elevated insulin-like growth factor 1 (IGF1) [75], which is a growth factor that drives cellular proliferation and survival and has thus been implicated in carcinogenesis of IGF-responsive tissues. Increased expression of IGF-1 and its cellular receptors are present in cancer tumours [76,77]. Our recent MR investigation demonstrated that genetically predicted IGF-1 was associated with increased risk of colorectal cancer, and, possibly, breast cancer, but not associated with overall or other site-specific cancers [78]. This suggests that the effect of height on cancer risk operates via pathways independent of IGF-1.

We observed a positive association between elevated genetically predicted BMI and overall cancer risk. This result replicates a recent MR investigation using 520 genetic variants for BMI in the UKBB that showed that overall cancer risk (excluding nonmelanoma skin cancer) and mortality was associated with elevated genetically predicted BMI. However, when dividing cancers into digestive system versus non-digestive system, genetically predicted BMI was only associated with digestive system cancers. This result has important clinical implications. Previous public health recommendations have advocated obesity as a generic risk factor for cancer prevention [79,80]. While our research supports a causal role of obesity in driving and protecting against certain cancers, it suggests differential effects of BMI and obesity in different malignancies, which should be explored further. A more nuanced message public health message with regard to obesity as a risk factor for digestive system cancers may be more appropriate.

Our study has several strengths. The MR design minimises the influence of environmental confounding factors and reverse causality, allowing for causal relationships to be better characterised. The UKBB is a large prospective cohort, allowing multiple cancer types to be studied in a single dataset and comparisons of estimates across cancers to be made. However, there are some limitations. The main limitation is the assumption that the genetic associations with cancer risk are mediated via the proposed risk factors. Additionally, estimates for some lower frequency cancer types are subject to low precision, and, therefore, results should be interpreted based on the magnitude of the associations rather than on *p*-values alone. Another shortcoming is that our findings may not be applicable to other ethnic groups as we confined the study population to individuals of European ancestries to minimise bias from population stratification. As we analysed middle- to early late-aged individuals, some cancers, particularly those with early onset and poor survival, may not be well captured in our analysis. Associations in the UKBB may be subject to selection bias [81], as participants in the UKBB are overall more healthy and better educated compared to the overall UK population [82]. Another potential source of bias is detection bias. The probability of diagnosis for less severe cancers (such as prostate cancer) may be more likely if the individual has comorbidities and so has more extensive contact with health services. While we wanted to perform analyses for specific cancer subtypes, we were unable to define these in a reliable way for the majority of site-specific cancers in the data available. Results for overall cancer are dependent on the characteristics of the analytic sample and the relative prevalence of different cancer types. In particular, cancers with greater survival chances will be overrepresented in the case sample. Selection of genetic variants was based on datasets that include the UKBB participants, and for FMI and FFMI, on a dataset comprised solely of the UKBB participants. This may lead to bias due to sample overlap and winner's curse. However, results were similar when genetic associations with cancers were taken from independent consortia. Our analysis assumes a linear relationship between the risk factors and the outcome. Quantitative estimates may be misleading if the true relationship is nonlinear, although estimates are still reflective of the presence and direction of the population-averaged causal effect [83]. As with all methodologies that aim to assess causal

relationships, MR makes untestable assumptions. The approach relies on the genetic associations with cancer risk being mediated via the body composition measures. While we were able to perform robust methods to assess sensitivity to this assumption, it remains a possibility that some genetic variants may influence cancer risk through other pathways than obesity. A further limitation is that the relationship between obesity and cancer risk may change over the life course. Typically, MR estimates reflect the impact of a lifelong difference in the trajectory of a risk factor, as they represent associations between genetically predicted levels of risk factors and outcomes. Finally, our study does not provide understanding of the physiological pathways by which obesity and height may affect carcinogenesis.

## Conclusions

In conclusion, this comprehensive MR study provides evidence that elevated BMI increases the risk of digestive system cancers, and BMI increases the risk of uterine cancer but is protective for other sex-specific cancers, including breast and prostate. We showed that elevated genetically predicted FMI is associated with liver, lung, and pancreatic cancer, with FFMI inversely associated with breast cancer. In contrast, genetically predicted height was consistently positively associated with overall cancer and several site-specific cancers. These findings suggest that obesity and body composition have particular causal relevance to specific cancer types.

## Supporting information

**S1 Checklist. STROBE Statement.** STROBE, Strengthening the Reporting of Observational Studies in Epidemiology.
(DOCX)

**S1 Fig. Flowchart of participant exclusion criteria in the UKBB.** PCA, principal component analysis; SD, standard deviation; UKBB, UK Biobank.
(PDF)

**S2 Fig. Effect sizes (OR per 1 kg/m$^2$ increase in risk factor) that can be detected for BMI, FMI, and FFMI with different number of cases in analyses based on 367,561 individuals, a significance level of 0.05, and a phenotypic variance of 4.05% for BMI, 3.15% for FMI, and 2.27% for FFMI.** As power calculators are not available for multivariable MR, all calculations are performed for univariable MR analyses based on each risk factor in turn. BMI, body mass index; FFMI, fat-free mass index; FMI, fat mass index; MR, mendelian randomisation; OR, odds ratio.
(PDF)

**S3 Fig. Associations of genetically predicted BMI with overall and site-specific cancers in the UKBB excluding outcomes that were self-reported only.** ORs are expressed per 1 kg/m$^2$ increase in BMI. Results are obtained from the random-effects inverse-variance weighted method. BMI, body mass index; CI, confidence interval; OR, odds ratio; UKBB, UK Biobank.
(PDF)

**S1 Table. Sources and definition of cancers in the UKBB. UKBB, UK Biobank.**
(PDF)

**S2 Table. SNPs used in the analyses of BMI. BMI, body mass index; SNP, single nucleotide polymorphism.**
(PDF)

**S3 Table. SNPs used as instrumental variables in the multivariable MR analyses of fat mass and fat-free mass indices.** MR, mendelian randomisation; SNP, single nucleotide polymorphism.
(PDF)

**S4 Table. SNPs used in the analyses of height. SNP, single nucleotide polymorphism.**
(PDF)

**S5 Table. Supplementary analyses of the association between genetically predicted BMI (per 1 kg/m$^2$ increase) and cancer.** BMI, body mass index.
(PDF)

**S6 Table. Supplementary analyses of the association between genetically predicted height (per 1 standard deviation increase) and cancer**
(PDF)

## Acknowledgments

The authors thank all investigators from the UK Biobank (UKBB), where data were conducted under application 29202.

**Disclaimers:** The views expressed are those of the authors and not necessarily those of the National Health Service, the National Institute for Health Research, or the Department of Health and Social Care.

## Author Contributions

**Conceptualization:** Mathew Vithayathil, Siddhartha Kar, Stephen Burgess.

**Data curation:** Paul Carter, Amy M. Mason, Stephen Burgess, Susanna C. Larsson.

**Formal analysis:** Siddhartha Kar, Amy M. Mason, Stephen Burgess, Susanna C. Larsson.

**Investigation:** Mathew Vithayathil, Stephen Burgess.

**Methodology:** Paul Carter, Amy M. Mason, Stephen Burgess, Susanna C. Larsson.

**Supervision:** Stephen Burgess.

**Writing – original draft:** Mathew Vithayathil, Susanna C. Larsson.

**Writing – review & editing:** Mathew Vithayathil, Paul Carter, Siddhartha Kar, Amy M. Mason, Stephen Burgess, Susanna C. Larsson.

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
