## [Editor Report · Decision Letter 0]

18 Aug 2020

Dear Dr Burgess, 

Thank you for submitting your manuscript entitled "Body size and composition and site-specific cancers in UK Biobank: a Mendelian randomisation study" for consideration by PLOS Medicine.

Your manuscript has now been evaluated by the PLOS Medicine editorial staff [as well as by an academic editor with relevant expertise] and I am writing to let you know that we would like to send your submission out for external peer review.

Kind regards,

Adya Misra, PhD,

Senior Editor

PLOS Medicine

---

## [Decision Letter · Decision Letter 1]

24 Sep 2020

Dear Dr. Burgess,

Thank you very much for submitting your manuscript "Body size and composition and site-specific cancers in UK Biobank: a Mendelian randomisation study" (PMEDICINE-D-20-03916R1) for consideration at PLOS Medicine. 

[LINK]

In light of these reviews, I am afraid that we will not be able to accept the manuscript for publication in the journal in its current form, but we would like to consider a revised version that addresses the reviewers' and editors' comments. Obviously we cannot make any decision about publication until we have seen the revised manuscript and your response, and we plan to seek re-review by one or more of the reviewers. 

We expect to receive your revised manuscript by Oct 15 2020 11:59PM. Please email us (plosmedicine@plos.org) if you have any questions or concerns.

We look forward to receiving your revised manuscript. 

Sincerely,

Adya Misra, PhD

Senior Editor 

PLOS Medicine

plosmedicine.org

Background- please provide brief context for this study, include 1-2 sentence of why this work may be important

Please structure the abstract to: background, methods and findings, conclusions

Please provide participant demographics 

Please provide additional details about the analyses carried out in the methods and findings section. 

In the last sentence of the Abstract Methods and Findings section, please describe the main limitation(s) of the study's methodology.

Please add p values along with 95% CI as needed

Please temper conclusions by adding “our results show” or similar 

Please ensure that the study is reported according to the STROBE guideline, and include the completed STROBE checklist as Supporting Information. Please add the following statement, or similar, to the Methods: "This study is reported as per the Strengthening the Reporting of Observational Studies in Epidemiology (STROBE) guideline (S1 Checklist)."

Did your study have a prospective protocol or analysis plan? Please state this (either way) early in the Methods section.

Please avoid using words such as “obese” and “overweight” throughout the text. Please amend to “with obesity” and “with overweight” throughout in line with people first principles. 

Please add a space between text and reference brackets throughout 

Methods- please provide further details on the data sources used, including names of registries or sources of electronic medical records

Table 1 please add units to the row “Age”

Comments from the reviewers:

Reviewer #1: In the present study, the authors analyzed the effect of body size and composition on cancer risk using UK Biobank data. Even though a wide-angled MR investigation was performed to evaluate the impact of obesity on cancer outcomes, there are several deficiencies that need to be clarified by the authors, amongst which some of these issues limited the interpretability of the findings. 

1. In the methods section, it was unclear how the quality control was performed. A clearer flowchart was needed, at least, in the supplementary. 

2. In the statistical analysis, it is not clear what kind of Mendelian randomized analysis was adopted in this study? And which R package is used for this analysis?

3. Because a recent Mendelian randomized study has shown a J shaped relation between BMI and all-cause mortality, is there a non-linear association between BMI and cancer risk? If so, will it affect the results of this study?

4. "The correlation of BMI with FMI was 0.84, and with FFMI was 0.66. The correlation between FMI and FFMI was 0.14." in this study, whether the instrumental variable of BMI is independent of FMI? Does the relation among those variables affect the findings of this study? 

5. Could the authors explain why there was a so strong effect of fat mass index on the stomach risk (OR=4.23) and liver cancer (OR=4.28)?

6. The incident events for most cancers were too small to perform a genetic analysis, as a result, the associations was not robust as far as I can see.

7. How about the direct and indirect effects of genetic components, BMI/ FMI/ FFMI, and cancer risk?

8. The number of overall cancer seems inconsistent with cancer-specific events according to the figures. 

Reviewer #2: ºWhat are the main claims of the paper and how significant are they for the discipline?

This study applied mendelian randomisation to explore the causal impact of BMI and height on all cancer, and site specific cancers. In addition to BMI the study also considers the impact of facets of BMI - fat and fat-free mass index. 

The main finding is that higher BMI has a causal but not consistent role in some site specific cancers e.g. increasing the risk of some, and protective for others. There was no evidence for a causal risk for overall cancer risk, contrary to previous reports (see following). In line with previous reports they found that genetically predicted increases in height is causally associated with increased rates of all cancers, and the majority of site specific cancers.

Significance for the discipline: This heterogeneity of direction with respects to BMI and site specific cancers is interesting but caution should be made in suggesting that as a result nuance should exist in regards to the obesity as a risk factor for specific cancers (discussion, third to last paragraph). Setting aside that a similar study using UK Biobank data (see following) found there was a consistent increase in risk for all cancer and all-cancer mortality for a genetically predicted increase in BMI (and thus reducing BMI would be expected to reduce the overall cancer burden in the population even if some subsets might rise), if the true causal estimate is that higher BMI does not increase overall risk for cancer as reported here, within the site specific cancers the protective effect from higher BMI would have to be balanced against both the increase in other cancers, and other BMI related morbidities. That is, the likely health advice would be to reduce BMI even in populations at risk of cancers where BMI may be protective. This is especially true given two of the cancers with reported reduced odds with higher BMI (melanoma and prostate) are either relatively less likely to kill compared to those that increase in risk with higher BMI, and/or have a range of interventions that could counter the potential increased risk (e.g. increased use of sunscreen for melanoma). That is, the advice would likely still be to reduce BMI in concert with appropriate cancer specific risk reduction strategies. Even for cancers with more severe prognosis, or fewer protective strategies (e.g. breast) the likely answer would be increased screening in those with genetically predicted higher BMI and still reduce BMI.

Action - The discussion should better explain what role nuance would play in BMI messaging in light of this paper

ºAre the claims properly placed in the context of the previous literature? Have the authors treated the literature fairly?

There are some important omissions in the reported literature that impact both the perceived novelty of this work, and its interpretation.

Gharahkani 2019 PMID: 30733581 performs a very similar analysis in the same UK Biobank dataset, using a larger BMI instrument to test both site specific and overall cancer risk. There are differences in the new analysis, but these should be interpreted with respect to that publication. This is especially relevant as this study found obesity is associated with all cancer, unlike this analysis, and this point has relevance with respects to the papers discussion.

While not significant at P< 0.05 Levy 2017 PMID: 28804972 reported an OR of 0.848 (0.658-1.093) for testicular cancer given a 1sd increase in BMI; while this is smaller than the causal estimate in this paper especially given the degree of change in BMI the Cis are wide for this estimate and likely overlap with the estimate and this should be discussed in the paper with respects to novelty

The discussion with respect to a positive association between height and overall cancer risk, and site specific cancer, while citing a similar paper (Ong et al 2018, ref 57) for specific cancers, does not note that ref 57 performed a very similar analysis in the same UK Biobank dataset and found consistent results; this should be addressed/contrasted in terms of novelty and what this study adds/improves.

The observation for BMI on melanoma is at odds with the uncited Dusingize 2020 PMID: 32068838 and this difference should be addressed; Dusingize 2020 also investigated height and found a consistent result for melanoma as this study.

Action: Cite these papers, and check for any additional relevant papers not reported here (I may have missed others) and then address them in their publication with respects to novelty, consistency and differences/advances.

ºDo the data and analyses fully support the claims? If not, what other evidence is required?

Partially:

More powerful BMI and height instruments are available using UK Biobank data or more recent GWAS meta-analyses e.g. Yengo 2018 PMID: 30124842 reports potential instruments explaining 6% of BMI and 24% of height variance. Gharahkani 2019 PMID: 30733581 constructed two instruments for BMI using cancer free UK Biobank participants that explained 4 or 7% of BMI variance. This would increase power for site specific estimates compared to those generated using their current instrument, which explains 1.6% of trait variance. While their power for height is greater (and arguably already sufficient), likewise Yengo 2018 or UK Biobank provides even more powerful instruments for height. 

Action: The authors should take advantage of data published or in hand to construct more powerful BMI/height instruments to improve precision/power.

The overall cancer measure appears to include non-melanoma cancers (keratinocyte cancers, basal and squamous cell carcinoma) as the overall cancer N is ~20k more than individual site count, and there is ~ this many ICD C44 cancer cases in UK Biobank. However table S1 does not report if C44 is included in the overall cancer phenotype. Observational studies suggest an inverse relationship between BMI and non-melanoma skin cancer (e.g. Zhou 2016 PMID: 27898109); if this is a causal effect, given how common non-melanoma skin cancers are in UK Biobank, this may explain why in this study BMI was associated with no increase in all-cancer while Gharahkani 2019 PMID: 30733581, which excluded these cancers, did. That is, the overall cancer estimate here may (crudely) be the sum of a protective effect on non-melanoma skin cancer, and in aggregate an increase for all other cancers.

This is an important distinction as non-melanoma skin cancers are very rarely lethal, are in general easily treated, and adequately controlled by sun smart campaigns - that is, even if BMI is (causally) protective for non-melanoma skin cancer such that the total number of cases of cancer does not change as BMI increases, trading off higher rates of NMSC for all other cancers is likely to be an acceptable decision.

Action: the impact of including, or excluding, non-melanoma skin cancers in all cancer estimates should be reported.

Action/comment: For the discussion hypothesis that increased IGF1 expression may underlie the causal effect of height - is this amendable to analysis with summary statistic data-based Mendelian randomization (SMR)? GTex reports eQTLs for this gene

ºPLOS Medicine encourages authors to publish detailed methods as supporting information online. Do any particular methods used in the manuscript warrant such publication? If a protocol is already provided, for example for a randomized controlled trial, are there any important deviations from it? If so, have the authors explained adequately why the deviations occurred?

Methods are sufficient barring other comments.

°Is this paper outstanding in its discipline? If yes, what makes it outstanding? If not, why not?

The paper is interesting but given the overlap with previous work is not in its current form outstanding.

ºDoes the study conform to any relevant guidelines such as CONSORT, MIAME, QUORUM, STROBE, and the Fort Lauderdale agreement?

N/A

ºAre details of the methodology sufficient to allow the experiments to be reproduced?

Action: related to above the methods should clarify how the overall cancer phenotype was constructed (e.g. does it include additonal cancers/ICD codes to those reported in sup table 1

ºIs any software created by the authors freely available?

N/A

ºIs the manuscript well organized and written clearly enough to be accessible to non-specialists?

In general the paper is well written and clear; two minor points:

1) Results paragraph 1 final sentences - are the reported correlations phenotypic, genetic etc? 

2) Results sub section BMI and cancer risk - the CIs for prostate cancer in line 4 don't match the figure/P-value. 

Reviewer #3: The MR analyses run in the paper are pretty standard. I have a few comments on the Mendelian Randomization analyses that the authors have performed:

1. One concern I have is that GWAS data for all the traits involved (the three risk factors and various cancer traits) are using the same UK Biobank data. A consequence is that for the exposure and outcome, some individuals are shared thus the summary statistics are also correlated due to non-heritable factors. In other words, for any specific SNP, the estimation of its effect on the exposure and outcome are not independent, which may bring bias to the MR analysis.

2. Related the question 1, I think to make the authors' conclusion more convincing, they can check whether than see replicable effects of BMI/FMI/height on cancers using BMI/FMI and cancer GWAS data from other cohorts (either European ancestry or other ethics).

3. In the discussions, the authors explained why BMI/height can have a causal effect on certain cancers. For height, the authors explained that its effect on height can be mediated by IGF1. I'm wondering if there are any GWAS data available for IGF1 (or similar traits) so that MR analyses can be performed to support this explanation. In general, I think the authors may need to provide more evidence to support their conclusions instead of using only the UK Biobank data and MR.

4. Since the authors look across about 20 cancers, I think multiple testing adjustment are needed. Since the significant p-values are not that small, I'm wondering if there are still many interesting signals left after adjustment. 

5. I recommend also performing MR reversing the role of cancer and BMI/height to see whether there is reserve causation (especially for BMI/FMI/FFMI).

6. In the section of "statistical analysis", the authors stated that "We performed sensitivity analyses using weighted median and MR-Egger", however I don't think weighted median and MR-Egger are sensitivity analyses, these are just two other MR methods. 

Reviewer #4: It was a pleasure to comment on this well-conducted and concisely written study. My main concern is limited reference to the most recent literature on adiposity and cancer; in particular, meta-analyses/umbrella reviews of observational studies.

Background

Because the focus is on obesity-related traits and cancer, I don't see the necessity to state that obesity is a risk factor for cardiovascular, liver, and musculoskeletal diseases.

I suggest shortening the citation of existing MR studies on body fatness and cancer and instead highlight the available evidence from observational studies. Based on grading by the World Cancer Research Fund and American Cancer Institute, strong evidence from meta-analyses of observational studies exists for the BMI and waist circumference and increased risk of postmenopausal, colorectum, endometrium, ovary, kidney, liver, gall bladder, stomach, esophagus, and pancreas, and moderate evidence exists for an association with cancers of the mouth, pharynx, larynx, prostate (advanced), male breast, and diffuse large B-cell lymphoma. Consider citing the most current grading of observational evidence from WCRFI/AICR/IARC [1-3] and umbrella reviews (e.g., [4-6]).

Could you be more specific which biases plaque observational research on obesity and cancer? Key biases include confounding by smoking [7-9] and reverse causation by subclinical and prevalent disease (cancer, cvd)[7, 10-12].

Methods 

Since adjustments (e.g. for waist in a GWAS of BMI) can bias the MR estimate, please report whether exposure GWAS adjusted for covariables [13-15].

There's a larger GWAS[16] on BMI than the GWAS by Locke - I am sure if summary data excluding UKBB is available. Reliance on BMI as a measure of general adiposity is limited by its inability to discriminate fat mass and lean body mass [12] and this can be more problematic in subclinical diseased individuals who often experience unintentional weight loss. I, therefore, welcome the use of fat mass and fat-free mass as additional exposures.

I doubt the validity of defining cancer cases based on 'self-report validated by a nurse', especially for less common cancers and histotypes. Can the authors provide data on validity? I suggest limiting outcome definition to cancer registry, outpatient and inpatient records, and death certificates.

I assume that using outcome data from the largest available GWAS meta-analyses would provide more cases and increase power. For example, a recent breast cancer GWAS[17] had 122,977 cases of European descent (another one[18] had 133,384 cases), compared to 13,666 cases included in the present MR study. Likewise, the largest available GWAS on lung carcinoma[19, 20] contains 10 times the number of cancer than the UK Biobank. I suggest adding replication analyses using the largest available cancer GWAS data. This would also address and circumvent possible biased introduced by weak instrumentation in 1-sample MR, winner's-curse (fmi,ffmi), and sample overlap. Although this could introduce new bias that should be discussed (e.g.,selection/survival bias). Can endometrial cancer be added as an outcome?

The 'overall cancer' outcome lumps together obesity-related cancers with cancers that are not affected by body fatness. I suggest omitting the total cancer analyses.

MR-Egger is sensitive to influential points in the regression[21] and point estimates are often imprecise[22, 23], as can be witnessed in Table S4. MR Egger is therefore not generally recommended as a preferred sensitivity analysis method [23-25]. I suggest to follow Slob and Burgess and report one of the three robust MR method classes discussed in [23] and omit MR Egger estimates. Consider adding E-values [26], variant-outcome associations (fewer assumptions [27, 28]), and negative controls [29].

Discussion

MR studies can, like observational studies, be subject to bias (pleiotropy - in particular when the biology of the variant-exposure association is ill-defined, population stratification, selection bias etc). Applying different study designs and analytical approaches may in the long run converge to provide triangulating evidence [30-32] and help strengthen answering causal relationships. I suggest separating the evidence from observational and MR studies; followed by a "grading" of the combined observational and MR evidence. It would be helpful to first summarize and grade the observational evidence based on the best available meta-analyses/umbrella reviews, possibly adapting the extensive work of WCRFI/AICR/IARC, then raise the limitations of observational studies (i..e., potential biases introduced by smoking and reverse causation by pre-existing disease). Then summarize the available MR-studies and illustrate where findings of observational and MR studies converge or diverge. Also, it would help to provide more details on the strengths and limitations of the available MR studies on obesity-traits and cancer. In particular, the number of cases or power achieved by the present MR study and previous MR studies should be discussed. Several recent large MR studies [33-37] that are not cited.

For breast cancer, according to observational research, being overweight or obese as an adult before menopause decreases the risk of premenopausal cancer of the female breast, but greater weight gain in adulthood increases the risk of postmenopausal breast cancer [1]. Observational research has provided the strongest evidence for obesity and postmenopausal [6]. The differential effects on pre- and postmenopausal breast cancer require further investigation A recent MR [36] study found that childhood adiposity might protect against adult breast cancer but did not consider breast cancer age of onset. Body composition is s time-varying exposure and the difficulties of MR to handle such exposures should be more clearly highlighted.

1. World Cancer Research Fund International and American Institute for Cancer Research, Diet, nutrition, physical activity and cancer: a global perspective. third expert report. 2018.

2. Wild, C.P., E. Weiderpass, and B.W. Stewart, World cancer report 2020. 2020, Lyon, France: International Agency for Research on Cancer.

3. Chan, D.S.M., et al., World Cancer Research Fund International: Continuous Update Project-systematic literature review and meta-analysis of observational cohort studies on physical activity, sedentary behavior, adiposity, and weight change and breast cancer risk. Cancer Causes Control, 2019. 30(11): p. 1183-1200.

4. Kyrgiou, M., et al., Adiposity and cancer at major anatomical sites: umbrella review of the literature. Bmj, 2017. 356: p. j477.

5. Raglan, O., et al., Risk factors for endometrial cancer: An umbrella review of the literature. Int J Cancer, 2019. 145(7): p. 1719-1730.

6. Kalliala, I., et al., Obesity and gynaecological and obstetric conditions: umbrella review of the literature. Bmj, 2017. 359: p. j4511.

7. Arnold, M., A.G. Renehan, and G.A. Colditz, Excess Weight as a Risk Factor Common to Many Cancer Sites: Words of Caution when Interpreting Meta-analytic Evidence. Cancer Epidemiol Biomarkers Prev, 2017. 26(5): p. 663-665.

8. Samet, J.M., Lung Cancer, Smoking, and Obesity: It's Complicated. J Natl Cancer Inst, 2018. 110(8): p. 795-796.

9. Song, M. and E. Giovannucci, Estimating the Influence of Obesity on Cancer Risk: Stratification by Smoking Is Critical. J Clin Oncol, 2016. 34(27): p. 3237-9.

10. Danaei, G., et al., Weight Loss and Coronary Heart Disease: Sensitivity Analysis for Unmeasured Confounding by Undiagnosed Disease. Epidemiology, 2016. 27(2): p. 302-10.

11. Flegal, K.M., B.I. Graubard, and S.W. Yi, Comparative effects of the restriction method in two large observational studies of body mass index and mortality among adults. European journal of clinical investigation, 2017. 47(6): p. 415-421.

12. Lee, D.H. and E.L. Giovannucci, The Obesity Paradox in Cancer: Epidemiologic Insights and Perspectives. Curr Nutr Rep, 2019. 8(3): p. 175-181.

13. Hartwig, F.P., et al., Bias in two-sample Mendelian randomization by using covariable-adjusted summary associations. BioRxiv, 2019: p. 816363.

14. Holmes, M.V. and G. Davey Smith, Problems in interpreting and using GWAS of conditional phenotypes illustrated by 'alcohol GWAS'. Mol Psychiatry, 2019. 24(2): p. 167-168.

15. Tan, V.Y., et al., Letter regarding article, "Associations of obesity and circulating insulin and glucose with breast cancer risk: a Mendelian randomization analysis". Int J Epidemiol, 2019. 48(3): p. 1014-1015.

16. Pulit, S.L., et al., Meta-analysis of genome-wide association studies for body fat distribution in 694 649 individuals of European ancestry. Hum Mol Genet, 2019. 28(1): p. 166-174.

17. Michailidou, K., et al., Association analysis identifies 65 new breast cancer risk loci. Nature, 2017. 551(7678): p. 92-94.

18. Zhang, H., et al., Genome-wide association study identifies 32 novel breast cancer susceptibility loci from overall and subtype-specific analyses. Nat Genet, 2020. 52(6): p. 572-581.

19. McKay, J.D., et al., Large-scale association analysis identifies new lung cancer susceptibility loci and heterogeneity in genetic susceptibility across histological subtypes. Nat Genet, 2017. 49(7): p. 1126-1132.

20. Wang, Y., et al., Association Analysis of Driver Gene-Related Genetic Variants Identified Novel Lung Cancer Susceptibility Loci with 20,871 Lung Cancer Cases and 15,971 Controls. Cancer Epidemiol Biomarkers Prev, 2020. 29(7): p. 1423-1429.

21. Burgess, S. and S.G. Thompson, Interpreting findings from Mendelian randomization using the MR-Egger method. Eur J Epidemiol, 2017. 32(5): p. 377-389.

22. Bowden, J., et al., Assessing the suitability of summary data for two-sample Mendelian randomization analyses using MR-Egger regression: the role of the I2 statistic. Int J Epidemiol, 2016. 45(6): p. 1961-1974.

23. Slob, E.A. and S. Burgess, A comparison of robust Mendelian randomization methods using summary data. Genet Epidemiol, 2020. 20: p. 1-17.

24. Burgess, S., C.N. Foley, and V. Zuber, Inferring Causal Relationships between Risk Factors and Outcomes Using Genetic Variation. Handbook of Statistical Genomics: Two Volume Set, 2019: p. 651-20.

25. Burgess, S., et al., Guidelines for performing Mendelian randomization investigations. Wellcome Open Research, 2020. 4(186): p. 186.

26. Swanson, S.A. and T.J. VanderWeele, E-Values for Mendelian Randomization. Epidemiology, 2020. 31(3): p. e23-e24.

27. VanderWeele, T.J., et al., Methodological challenges in mendelian randomization. Epidemiology, 2014. 25(3): p. 427-35.

28. Didelez, V. and N. Sheehan, Mendelian randomization as an instrumental variable approach to causal inference. Stat Methods Med Res, 2007. 16(4): p. 309-30.

29. Sanderson, E., et al., The use of negative control outcomes in Mendelian Randomisation to detect potential population stratification or selection bias. BioRxiv, 2020.

30. Mariosa, D., et al., Commentary: What can Mendelian randomization tell us about causes of cancer? Int J Epidemiol, 2019. 48(3): p. 816-821.

31. Mamluk, L., et al., Evidence of detrimental effects of prenatal alcohol exposure on offspring birthweight and neurodevelopment from a systematic review of quasi-experimental studies. Int J Epidemiol, 2020.

32. Lawlor, D.A., K. Tilling, and G. Davey Smith, Triangulation in aetiological epidemiology. Int J Epidemiol, 2016. 45(6): p. 1866-1886.

33. Shu, X., et al., Associations of obesity and circulating insulin and glucose with breast cancer risk: a Mendelian randomization analysis. Int J Epidemiol, 2019. 48(3): p. 795-806.

34. Langdon, R.J., et al., A Phenome-Wide Mendelian Randomization Study of Pancreatic Cancer Using Summary Genetic Data. Cancer Epidemiol Biomarkers Prev, 2019. 28(12): p. 2070-2078.

35. Cornish, A.J., et al., Modifiable pathways for colorectal cancer: a mendelian randomisation analysis. Lancet Gastroenterol Hepatol, 2020. 5(1): p. 55-62.

36. Richardson, T.G., et al., Use of genetic variation to separate the effects of early and later life adiposity on disease risk: mendelian randomisation study. Bmj, 2020. 369: p. m1203.

37. Yarmolinsky, J., et al., Appraising the role of previously reported risk factors in epithelial ovarian cancer risk: A Mendelian randomization analysis. PLoS Med, 2019. 16(8): p. e1002893.

[LINK]

---

## [Decision Letter · Decision Letter 2]

16 Jun 2021

Dear Dr. Burgess,

Thank you very much for re-submitting your manuscript "Body size and composition and risk of site-specific cancers in UK Biobank and large international consortia: a Mendelian randomisation study" (PMEDICINE-D-20-03916R2) for review by PLOS Medicine. We do apologize for the long delay in sending you a decision. 

I have discussed the paper with editorial colleagues and it was also seen again by three reviewers. I am pleased to tell you that, provided the remaining editorial and production issues are fully dealt with, we expect to be able to accept the paper for publication in the journal.

[LINK]

Please let me know if you have any questions, and we look forward to receiving the revised manuscript.   

Sincerely,

Richard Turner, PhD

rturner@plos.org

Requests from Editors:

Please finalize the data statement (submission form). 

Please correct "casual" late in the abstract and any other instances.

Please add bullet points to the individual points in the author summary.

We suggest removing the ORs and 95% CI from the author summary where these are also quoted in the abstract.

Regarding the final sentence of the introduction ("This study is important ..."), please amend this to begin "We aimed to elucidate ..." or similar, or move it to the Discussion. 

Please remove the "Role of the funding source" section from the Methods - information on funding will appear in the article metadata, via entries in the submission form. 

Similarly, please remove the information on competing interests and funding from the end of the main text.

Noting some of the figures, we generally ask that exact p values are quoted, or "p<0.001", unless there is a specific statistical reason for quoting smaller values. 

Please use the journal name abbreviation "PLoS ONE". 

Comments from Reviewers:

*** Reviewer #2: 

The authors have satisfactorily addressed my concerns.

*** Reviewer #3: 

The authors have addressed all my previous comments.

*** Reviewer #4: 

Thank you to the authors for responding to my previous comments. I was satisfied with all of their answers.

***

[LINK]

---

## [Editor Report · Decision Letter 3]

21 Jun 2021

Dear Dr Burgess, 

On behalf of my colleagues and our Academic Editor, Dr Law, I am pleased to inform you that we have agreed to publish your manuscript "Body size and composition and risk of site-specific cancers in UK Biobank and large international consortia: a Mendelian randomisation study" (PMEDICINE-D-20-03916R3) in PLOS Medicine.

PRESS

Sincerely, 

Richard Turner, PhD 

rturner@plos.org